# LOMAC: GNN-BASED DEEP REINFORCEMENT LEARNING WITH ONE-WAY MARKOV CHAIN FOR GRAPH COLORING

## ABSTRACT

The graph coloring problem (GCP) is an NP-hard combinatorial optimization task aimed at assigning the minimum number of colors to graph vertices such that no two adjacent vertices share the same color. While deep reinforcement learning (DRL) and graph neural networks (GNNs) are promising approaches to solving the GCP, their scalability is usually limited by the large number of Markov states and high computational complexity as the graph size increases. In this paper, we introduce LOMAC, a novel GNN-based DRL framework that integrates a one-way, two-dimensional Markov chain and a linear-complexity GNN model with pseudonode-enhanced message passing. This integration significantly reduces both space and computational complexity. We transform the GCP into a one-way Markov chain model, introducing two key concepts: Markov state potential and graph state potential. Through theoretical analysis of Markov- and graph-state potentials, we effectively guide the search for an optimal vertex-coloring solution. We show that LOMAC reduces the number of Markov states from $O(K^N)$ to $O(NK)$, simplifying decision-making with unidirectional state transitions. Additionally, an invalid action penalty mechanism is implemented to further optimize the coloring process. Experimental results in various sizes of *Erdős–Rényi-* and *Barabási–Albert* graphs and 16 real-world benchmarks demonstrate that LOMAC achieves state-of-the-art performance in the number of required colors.

## 1 INTRODUCTION

The graph coloring problem (GCP) is a critical challenge in combinatorial optimization (CO) and graph theory. It involves assigning the fewest number of colors to the vertices of a graph so that no two adjacent vertices share the same color. Efficient solutions to GCP have significant applications, including resource scheduling (Rina et al., 2022), register allocation (Das et al., 2020), pilot assignment (Liu et al., 2020), content caching (Javedankherad et al., 2022), and wireless channel assignment (Ge et al., 2023). However, determining whether a graph can be colored with $K$ colors is NP-complete, and minimizing the chromatic number is NP-hard. This means that there is no polynomial-time algorithm for solving the GCP under the $P \neq NP$ conjecture. Recent advances have explored deep reinforcement learning (DRL) and graph neural networks (GNN) (Colantonio et al., 2024; Pugachewa et al., 2024; Lemos et al., 2019; Langedal & Manne, 2024; Prates et al., 2019; Huang et al., 2019; Yuan et al.,

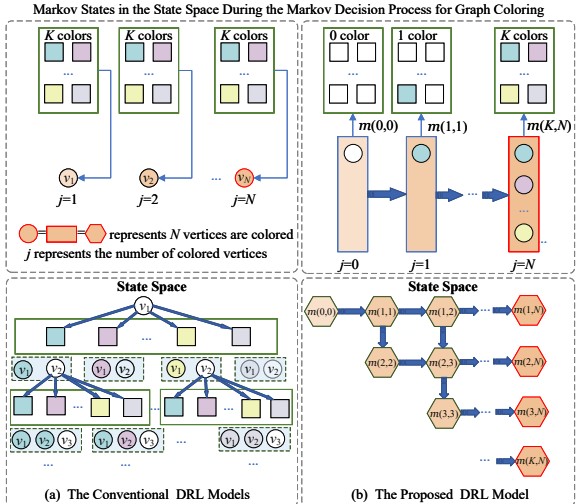

Figure 1: Comparison of Markov states in the state space during the Markov decision process for graph coloring between conventional DRL models (with $O(K^N)$ Markov states) and the proposed DRL model (with $O(NK)$ Markov states).

2024) as promising approaches to solve GCP. DRL aids in sequential decision making, while GNNs are capable of modeling and processing graph-structured data. However, these models face scalability issues due to the large number of Markov states in the state space, even for moderately sized graphs. This is because they typically model the entire coloring process as a Markov Decision Process (MDP), in which each possible color assignment for every node in the graph is treated as an independent Markov state. As the size of the graph increases, the state space expands exponentially.

To address the scalability limitations of traditional methods, we propose LOMAC, which integrates a Markov chain into DRL and employs a pseudonode-enhanced GNN for efficient graph coloring. Specifically, we design a one-way, two-dimensional Markov chain with finite states, which significantly reduces state space and computational demands compared to conventional DRL methods. As shown in Fig.1, conventional DRL models assign one of the $K$ colors to each vertex, resulting in $O(K^N)$ Markov states. In contrast, our model reduces the state space to $O(NK)$, which is at most $O(N^2)$ when the chromatic number is unknown. The one-way Markov chain restricts transitions to a single direction, further simplifying the decision-making process and reducing computational requirements. Additionally, we introduce new definitions of the Markov state potential and graph state potential. By analyzing the relations between these potentials, we establish inequality constraints that guide the coloring process toward states with fewer colors. We also propose a potential-based reward function that penalizes invalid actions and guides the identification of optimal coloring strategies. Inspired by the work of $N^2$ (Sun et al., 2024), we designed a pseudonode-enhanced GNN model for GCP that enables the passing of dynamic messages in linear time. This model utilizes pseudonodes as intermediaries for message passing to effectively learn Q-value embeddings for node selection actions. This design reduces computational overhead and alleviates dependence on the input graph topology. The experimental results demonstrate that LOMAC achieves better performance compared to existing methods. The contributions of this paper are threefold.

- We provide a GNN-based DRL solution to the GCP by introducing a novel one-way, two-dimensional Markov chain with finite states. This design significantly reduces the state space and computational complexity, even for graphs with unknown chromatic numbers. We also propose a pseudonode-enhanced GNN for linear-time message passing, which effectively learns Q-value embeddings for node selection actions.

- We introduce two new definitions, the Markov state potential and the graph state potential. Furthermore, we establish inequality relations between the potential values of Markov and graph states to find an optimal vertex-coloring solution and validate a reward function that enhances model efficiency and solution quality. Additionally, we propose an invalid action penalty mechanism to further optimize the coloring process.

- We show that LOMAC outperforms existing methods in various sizes of *Erdős–Rényi* (ER)- and *Barabási–Albert* (BA) graphs, as well as 16 real-world benchmarks, excelling in the number of required colors, matching ratio, and execution time.

## 2 RELATED WORKS

**Traditional Heuristic Algorithms.** Early heuristic approaches to graph coloring often employed greedy strategies such as Largest First (LF), Smallest First (SF) (Gebremedhin et al., 2013), and Tabu Search (Blochliger & Zufferey, 2008). More recent evolutionary algorithms, such as simulated annealing (Kose et al., 2017), heuristic feedback (Inaba et al., 2022), and genetic algorithms (Shem-Tov & Elyasaf, 2024), have also shown success. For example, Inaba *et al.* (Inaba et al., 2022) applied the Potts model to graph coloring, iteratively updating interaction matrices to minimize Potts energy. Although these methods provide feasible solutions, they are tremendously time consuming and often based on manually crafted heuristics, which limits their ability to explore the solution space more effectively.

**GNN Methods.** GNNs have become a popular approach for solving combinatorial optimization problems by learning features from graph-structured data (Kose et al., 2017; Prates et al., 2019; Lemos et al., 2019). Once trained, GNNs can efficiently generate solutions for new instances. For example, a Potts model inspired GNN (Colantonio et al., 2024) was applied to the graph coloring problem, while Pugachewa *et al.* (Pugachewa et al., 2024) used recurrent GNNs to obtain optimal solutions. Langedal *et al.* (Langedal & Manne, 2024) introduced a GNN-based ordering heuristic for

graph coloring, achieving execution times comparable to greedy algorithms. However, GNN-based models often require substantial training in various instances to be generalized effectively.

**DRL Methods.** Unlike GNN models, DRL provides a dynamic framework for learning optimal policies based on expected outcomes (Ma et al., 2020). For example, Li *et al.* (Li et al., 2021) developed an unsupervised DRL method for a traveling salesman problem. Zhang *et al.* (Zhang et al., 2022) proposed a meta-learning-based DRL model for handling multi-objective combinatorial optimization problems. Although DRL is superior in learning optimal policies from states and rewards, it struggles to fully leverage graph-structured data.

**GNN-based DRL Algorithms.** A notable advancement in this area is FastColorNet (Huang et al., 2019), a graph coloring algorithm that integrates DRL and GNN for vertex color assignments. Other frameworks, such as S2V-DQN (Khalil et al., 2017; Manchanda et al., 2020), ECO-DQN (Barrett et al., 2020), and those in (Xu et al., 2022; Li et al., 2023; Liu & Huang, 2023), model combinatorial optimization problems as MDP and use GNNs for representation learning to guide the actions of DRL agents. These approaches use graph embeddings as Q-values for each node and add nodes to the solution one by one, based on their corresponding Q-values. For large-scale GCPs, Yuan *et al.* (Yuan et al., 2024) proposed a multicolumn selection strategy combining DRL and GNN, significantly reducing training iterations and runtime. Despite these advancements compared to standalone GNN and DRL methods, GNN-based DRL algorithms still face scalability issues due to high space and computational complexity. In this paper, we introduce a novel GNN-based DRL framework that integrates a one-way, two-dimensional Markov chain with $O(NK)$ states and a linear-complexity GNN model, significantly reducing both space and computational complexity.

**LLM approaches.** Large Language Models (LLMs) have recently gained significant attention for logical reasoning and planning tasks. Recent studies (Stechly et al., 2023; Zhang et al., 2023; Zhou et al., 2023; Zhang et al., 2024; Mittal et al., 2024) have explored their potential for graph coloring using prompt-based reasoning techniques, such as chain-of-thought (Zhang et al., 2023) and least-to-most prompting (Zhou et al., 2023). For example, Stechly *et al.* (Stechly et al., 2023) investigated iterative prompting for graph coloring and found that self-critique-based prompting struggled with reasoning and correctness verification. Although these methods show promise in enhancing logical reasoning and solution accuracy, LLMs are inherently sequence-based, lacking explicit logical reasoning modules. This makes them prone to generating invalid or suboptimal solutions.

## 3 LOMAC FRAMEWORK

This section introduces the one-way Markov chain model for graph coloring, presents the input representation for the model, and describes the proposed framework.

### 3.1 ONE-WAY MARKOV CHAIN MODEL

Consider an undirected graph $G = \{V, E\}$, where $V$ represents the vertices and $E = \{(i, j)|i, j \in V\}$ represents the edges. The goal of graph coloring is to assign a unique color $c$ to each vertex $v$, minimizing the chromatic number while ensuring that adjacent vertices do not share the same color. Let $K$ denote the number of colors, and let $N$ be the number of vertices. The set of colors is represented as $C = \{c_1, c_2, \ldots, c_K\}$, with $c^v$ denoting the color assigned to the vertex $v$, and $\eta_v$ indicating the number of neighbors of $v$. $V^c$ represents the set of colored vertices. As shown in Fig. 2, we introduce a one-way, two-dimensional Markov chain to model the coloring process. Starting from an initial state $m(0, 0)$, where no vertex is colored, the process progresses to the state $m(1, 1)$ with one colored vertex, and continues up to $m(i, j)$, where $j$ vertices are colored using $i$ colors. A graph with $N$ vertices requires at most $N$ colors. The total number of states is not more than $O(N^2)$, especially in a fully connected graph, where each vertex requires a unique color.

In this model, transitioning from state $m(i, j)$ involves three scenarios: 1) If a valid color exists for vertex $v$, the state moves to $m(i, j + 1)$; 2) If no valid color exists for $v$, a new color is introduced and the state moves to $m(i + 1, j + 1)$;

3) If $v$ is already colored, the state remains at $m(i, j)$. The process reaches completion at the state $m(i, N)$, where all vertices are colored. Except for the third case, the model ensures a streamlined one-way transition towards the final state. Importantly, Markov states and graph states hereinafter are distinct. Markov state is determined by the number of colored vertices and the colors used, while the graph state is defined by the coloring of vertices. It is easy to see that the graph state $s$ cannot be inferred from a Markov state $m(i, j)$, but the corresponding Markov state can be derived from the graph state $s$ by counting the colored vertices and the colors used. To avoid confusion, we will specify these states in subsequent sections.

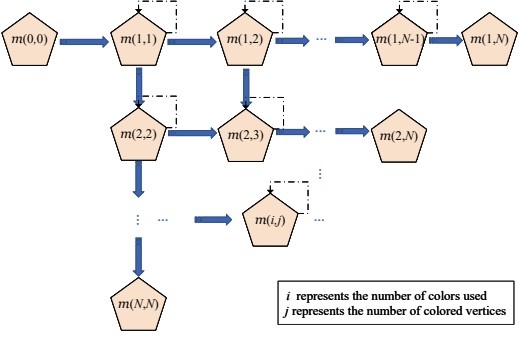

Figure 2: The one-way Markov chain model for graph coloring.

## 3.2 INPUT REPRESENTATION

To formalize the input of the proposed LOMAC framework, we introduce the following definitions relevant to the GCP.

**Definition 1 (Colored Edge and Uncolored Edge).** An edge is a colored edge if at least one of its incident vertices is colored. Otherwise, it is an uncolored edge.

**Definition 2 (Colored Degree of a Graph).** The colored degree $\zeta_G$ of a graph $G = \{V, E\}$ is the total number of colored edges in $E$. Proper coloring ensures that the adjacent vertices have different colors.

**Definition 3 (Uncolored Vertex Degree).** For an uncolored vertex $v$, its uncolored degree $\ell_v$ is the number of uncolored edges incident to $v$. For colored vertices, $\ell_v = 0$.

**Definition 4 (Uncolored Degree of a Graph).** The uncolored degree $\zeta'_G$ of graph $G = \{V, E\}$ is the total count of uncolored edges, with $\zeta_G + \zeta'_G = |E|$ representing the total number of edges.

**Definition 5 (Valid Color Set).** For an uncolored vertex $v$, the valid color set $C^v$ includes colors that ensure proper coloring when assigned to $v$. If $C^v = \varnothing$, a new color must be introduced.

**Definition 6 (Saturation Degree).** The saturation degree $\rho_v$ of a vertex $v$ is defined as the number of its colored neighbors.

**Definition 7 (Degree Centrality).** The centrality of the degree $d_v$ of a vertex $v$ is defined as the ratio of its degree to the maximum vertex degree in the graph.

**Definition 8 (Color Number of Neighbors).** The color number of neighbors $\delta_v$ is defined as the number of different colors that appear among its adjacent vertices.

We model the graph coloring task as an MDP. This process is defined as $M = (\mathcal{S}, \mathcal{A}, \mathcal{R}, \mathcal{V}, \gamma)$, where $\mathcal{S}$ is the graph state space (distinct from the Markov state space in Fig. 2), $\mathcal{A}$ is the action space, $\mathcal{R}$ is the reward function, $\mathcal{V}(s)$ is the state value of $s$ and $\gamma$ is the discount factor. A graph state $s$ is represented as $s = [s'_0, s'_1, \ldots, s'_N] \in \mathbb{R}^{N \times m'}$, where $s'_i$ is the set of attributes for the vertex $v_i$, which contains $m'$ attributes. Specifically, the state of each vertex is represented by six attributes: its color, uncolored vertex degree, size of the valid color set, saturation degree, degree centrality, and color number of neighbors. Initially, all vertices are uncolored, so $c^{v_i} = -1$ for all $1 \leq i \leq N$. The uncolored degree $\ell_v$ is updated when the neighboring vertices are colored. The action space $\mathcal{A}$ consists of actions $a_{v_i}$ for each vertex $v_i$, and $a_{v_i}$ represents the selection and coloring of the vertex $v_i$. Only one vertex is colored per step, and once a vertex is colored, it is not recolored. If one vertex $v_j$ has been a colored vertex, the action $a_{v_j}$, i.e., selecting and coloring $v_i$ repeatedly, is called an invalid action. We use deep Q-learning and a message passing neural network to solve the graph coloring task. For further details on deep Q-learning and message-passing neural networks, please refer to Appendix A. The action that maximizes the Q-value is selected at each step:

$$a^* = v^* = arg \max_{a \in \mathcal{A}} Q(s, a). \tag{1}$$

To optimize the coloring process while minimizing the number of colors used, we establish the following inequality relations between the potentials of Markov states (denoted as $V'_{i,j}$ for the Markov

state $m(i, j)$:

$$\begin{cases} V'_{i,j+1} - V'_{i,j} \geq 1. \\ V'_{i,j} = V'_{i+1,j+1}. \\ V'_{i,j} > V'_{i+1,j}. \end{cases} \tag{2}$$

As previously outlined, the agent's progression from the current state is restricted to transitions to the right or downward, without considering invalid actions. This restriction ensures that the potential of Markov states increases monotonically, as specified by the first two equations. Eq. (2) shows that the Markov state of coloring $j$ vertices with fewer colors has greater potential. To quantify the potential of a graph state $s$, we define it as a combination of $V'_{i,j}$, which represents the potential of the Markov state, and $V''(\zeta_G)$, the ratio of colored edges within the graph. This is formalized as follows.

$$\begin{cases} \mathcal{V}(s) = V'_{i,j} + V''(\zeta_G), \\ V''(\zeta_G) = \frac{\zeta_G}{\|E\|}, \end{cases} \tag{3}$$

where $V''(\zeta_G)$ is the proportion of colored edges to the total number of edges. This formulation reveals that $\mathcal{V}(s)$ depends on the number of colors, colored vertices, and colored edges within the graph $G$. In particular, different graph states may share identical counts of colors, colored vertices, and colored edges.

**Theorem 1**. For the Markov decision process $M = (\mathcal{S}, \mathcal{A}, \mathcal{R}, \mathcal{V}, \gamma)$, the potential of the Markov and graph states adheres to Eqs. (2) and (3), respectively, producing the following:

$$\begin{cases} \mathcal{V}(s(i, n)) \geq \mathcal{V}(s(i', n)) \text{ if } i' \leq i. \\ \mathcal{V}(s(i, j)) \geq \mathcal{V}(s(i, j+1)). \\ \mathcal{V}(s(i, j)) \leq \mathcal{V}(s(i+1, j+1)). \\ \mathcal{V}(s(i, j)) \geq \mathcal{V}(s(i+1, j)). \end{cases} \tag{4}$$

*Proof.* Please refer to Appendix B.

**Theorem 2**. For MDP $M = (\mathcal{S}, \mathcal{A}, \mathcal{R}, \mathcal{V}, \gamma)$ as shown in Fig. 2, let $\mathcal{V}(s(k, N))$ be the largest potential of the graph states when all $N$ vertices are colored. Here, $k$ equals the chromatic number $\kappa$.
*Proof.* Please refer to Appendix C.

The proofs for these statements demonstrate the inherent monotonic increase in the potential of graph states and Markov states within the MDP model, highlighting the model's preference for graph states with fewer colors and more colored edges, given the same number of colored vertices.

## 3.3 Constrained Reward Shaping

In this subsection, we introduce a graph state potential-based reward function that also penalizes invalid actions. Based on the Markov chain model described previously, there are three types of actions $a_v$, each corresponding to a specific state transition: from state $s(i, j)$ to $s(i, j+1)$, from $s(i, j)$ to $s(i+1, j+1)$, and from $s(i, j)$ to $s(i, j)$. $s(i, j)$ denotes a graph state in which $j$ vertices are colored using $i$ colors. It differs from the abstract Markov state $m(i, j)$ by preserving the full vertex-level coloring configuration. For the first two types of action, the reward is the potential difference between the old and new graph states. For the invalid action, we assign a negative constant $z$ (where $z < 0$) as a penalty. The reward function and the corresponding graph state potential function are defined below.

$$\mathcal{R}(s, a_v) = \begin{cases} 1 + \frac{\Delta\zeta_G}{\|E\|}, & c^v = -1 \wedge \mathcal{C}^v \neq \emptyset \\ \frac{\Delta\zeta_G}{\|E\|}, & c^v = -1 \wedge \mathcal{C}^v = \emptyset \\ z, & c^v \neq -1 \end{cases} \tag{5}$$

$$\mathcal{V}(s(i, j)) = (j - i) + \frac{\zeta_G^{(i,j)}}{\|E\|} \tag{6}$$

where $\Delta\zeta_G$ denotes the change in $\zeta_G$ induced by action $a_v$, and $\zeta_G^{(i,j)}$ represents the number of colored edges in the Markov state $s(i, j)$.

## 3.4 SYSTEM ARCHITECTURE

As illustrated in Fig. 3, the proposed LOMAC system architecture consists of three phases: 1) GNN-based decision making phase: Identifies the optimal vertex for coloring. 2) Color assignment phase: Dynamically assigns colors and updates the graph state. 3) Prioritized experience replay phase: Consolidates learning trajectories for model refinement.

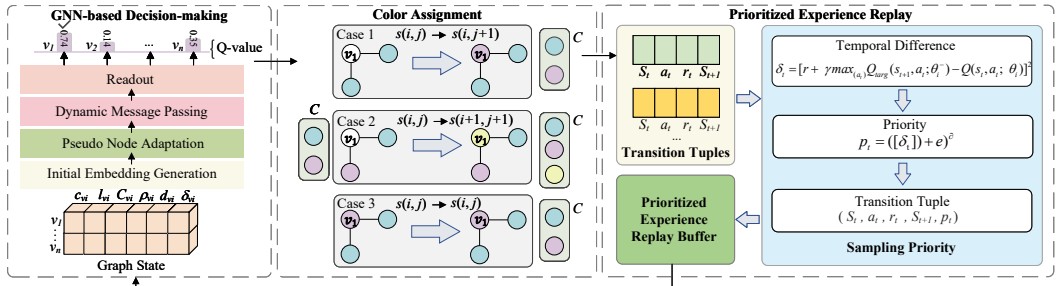

Figure 3: GNN-based DRL framework for GCPs, consisting of three phases: GNN-based decision-making, color assignment, and prioritized experience replay.

**GNN-based decision making phase.** The network processes the graph state $s = [s'_1, \ldots, s'_N] \in \mathbb{R}^{N \times 6}$, where each node $v_i$ is represented by $s'_i = [c^{v_i}, \ell_{v_i}, C^{v_i}, \rho_v, d_v, \delta_v]$. Using the GNN model described in Section 4, which includes initial embedding generation, pseudonode adaptation, dynamic message passing, and readout blocks, the optimal vertex for coloring is selected as $a^* = v^* = \arg\max_{a \in \mathcal{A}} Q(s, a)$.

**Color assignment phase.** This phase assigns an appropriate color to the selected vertex, with the color assignment process detailed in Section 3.1.

**Prioritized experience replay phase.** To improve training stability, we use a prioritized experience replay to focus on transitions with higher temporal difference errors. At each step, the agent stores the transition tuple $(s_t, a_t, r_t, s_{t+1}, p_t)$ in a replay buffer, where the priority $p_t$ is computed as:

$$p_t = \delta_t + \epsilon. \tag{7}$$

The temporal difference (TD) error is defined as:

$$\delta_t = \left[ r_t + \gamma \max_{(a_t)} Q_{\text{targ}}(s_{t+1}, a_t; \theta_i^-) - Q(s_t, a_t; \theta_i) \right]^2, \tag{8}$$

where $s_t$ is the current state, $a_t$ is the action taken at state $a_t$, $r_t$ is the immediate reward received after executing $a_t$, and $s_{t+1}$ is the next state reached. $\gamma$ is the discount factor that controls future reward weighting, $\epsilon$ is a small positive constant to ensure a nonzero sampling probability, $\theta_i$ denotes the parameters of the current Q network, $\theta_i^-$ denotes the parameters of the target Q network, and $Q_{\text{targ}}$ denotes the target Q network which is a slower-updating copy of the current Q network. The training and testing processes of LOMAC are described in Appendix E.

## 4 SYSTEM IMPLEMENTATION OF LOMAC

This section details the implementation of the core component in LOMAC, GNN-based decision-making using pseudonode-enhanced message passing. The GNN architecture establishes a shared embedding space $\mathbf{H} \in \mathbb{R}^q$ for both physical graph nodes $V = \{v_i\}_{i=1}^N$ and trainable pseudonodes $U = \{u_j\}_{j=1}^M$. Let $\mathbf{Q} \in \mathbb{R}^{N \times q}$ and $\mathbf{R} \in \mathbb{R}^{M \times q}$ represent their respective state matrices. Node proximity is measured through adaptive feature correlation:

$$\phi(r_i, r_j) = \sum_{t=1}^q \lambda_t \sigma(r_{i,t}) \sigma(r_{j,t}), \tag{9}$$

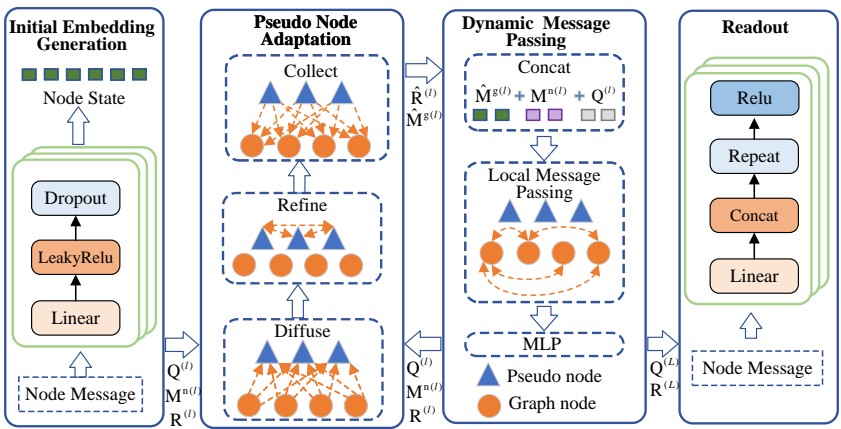

Figure 4: GNN-based decision making architecture featuring four key blocks, initial embedding generation, adaptive pseudo-node coordination, message passing (local/global), and readout.

where $\lambda_{1:q}$ are learned attention weights and $\sigma$ denotes a perceptron layer equipped with LeakyReLU activation and dropout regularization. The architecture employs four specialized processing blocks described in the following.

**Initial Embedding Generation**. The block establishes initial node representations through the projection of permutation-equivariant features.

$$\mathbf{Q}^{(0)} = f_\theta(\mathbf{M}_V^{(0)}) \in \mathbb{R}^{N \times q}, \tag{10}$$

where $f_\theta$ is a linear transformation learned initialized from the graph state $s$.

**Pseudo Node Adaptation**. The block orchestrates the global information flow through three phase-coordinated operations:

$$\text{Diffusion: } \mathbf{G} = \mathbf{E}^{np}\mathbf{M}_V^{(l-1)}, \; \mathbf{E}_{ij}^{np} = \phi(r_i, q_j) \tag{11}$$

$$\text{Refinement: } \hat{\mathbf{G}} = \mathbf{E}^{pp}\mathbf{G}, \; \mathbf{E}_{ij}^{pp} = \phi(r_i, r_j) \tag{12}$$

$$\text{Redistribution: } \hat{\mathbf{M}}^{g(l)} = \mathbf{E}^{pn}\mathbf{M}_U^{(l)}, \; \mathbf{E}_{ij}^{pn} = \phi(q_i, \hat{r}_j) \tag{13}$$

**Dynamic Message Passing**. The block combines neighborhood aggregation with global state diffusion. For each graph node $v$:

$$\mathbf{M}_v^{loc(l)} = \frac{1}{\eta_v + 1}\left[ m_u^{l-1} + \sum_{u' \in \mathcal{N}(v)} \sigma(m_{u'}^{l-1} \| m_u^{l-1}) \right] \tag{14}$$

$$\hat{\mathbf{Q}}^{(l)} = \mathbf{Q}^{(l-1)} + \sigma(\mathbf{M}^{loc(l)}) \tag{15}$$

Global message updates follow operator sequences similar to those in Eqs. (11)-(13) but using current-layer states.

**Readout**. The block computes the $Q$ value of the action $a_{v_i}$ after the $L$ propagation layers:

$$Q(a_{v_i}) = \psi(\mathbf{Q}^{(L)}[i, :]), \tag{16}$$

where $\psi$ maps the final node states to the $Q$ value.

## 5 EXPERIMENT

In this section, we evaluate the performance of LOMAC on synthetic and real-world benchmarks. LOMAC is implemented in PyTorch and is trained on a Nvidia GeForce RTX 4090 GPU. Detailed experimental settings are provided in Appendix D.2.

**Dataset.** We generate random graphs based on the ER and BA distributions. The model is trained on $W = 12500$ randomly generated graph samples and evaluated on a separate set of 100 holdout graphs drawn from the same distributions. We examine the experimental results for the sizes of the graphs $N = 20, 40, 60, 200, 500, 1000$. ER graphs are generated using the ER $G(N, p' = 0.15)$ model, while BA graphs follow the BA model, where each node is connected to $m_0 = 4$ nodes.

**Baseline algorithms.** We compare LOMAC with four categories of baseline algorithms: (1) Traditional heuristic algorithms, including Tabu (Blochliger & Zufferey, 2008) and DLF-GA (Gebremedhin et al., 2013); (2) GNN method, specifically GNN-GCP (Lemos et al., 2019); (3) GNN-based DRL algorithms, including SAT-DRL (Yolcu & Póczos, 2019), ECO-DQN (Barrett et al., 2020) and MCSS (Yuan et al., 2024); (4) LLMs, including LTMP (Zhou et al., 2023) and AUTO-COT (Zhang et al., 2023). For details of the baseline algorithms, please refer to Appendix D.1. We also performed Wilcoxon's significance tests to assess the statistical robustness of LOMAC against these baselines. The detailed results of the Wilcoxon test are reported in Appendix D.4.

**Evaluation Metrics.** We adopted the following evaluation metrics. Required number of colors (RNC): A smaller RNC indicates a more efficient coloring, which is crucial for applications such as registration allocation and pilot assignment. Matching ratio (MR): The proportion of test cases in which the model successfully matches the known chromatic number $\kappa$, obtained using the CSP-Solver[1] as reference. Execution time (ET): The average time taken to solve the coloring tasks, excluding model training time for learning-based algorithms, as the training is performed offline.

## 5.1 PERFORMANCE ON ER AND BA GRAPH INSTANCES

We compare the performance of LOMAC with eight baseline algorithms on the ER and BA graphs in terms of RNC, MR, and ET, averaged over 100 test samples, as shown in Table 1. Additional results are reported in Appendix D.5. Unsolvable instances are marked as 'NA'. From the tables, we observe that LOMAC outperforms other algorithms in RNC. As the size of the graph increases from $N = 40$ to 200 and 1000, LOMAC significantly reduces the number of colors required, while most algorithms fail to provide solutions for larger graphs. Even in the worst cases, the LOMAC RNC deviates by no more than 4% from the best results in multiple runs. Furthermore, LOMAC strikes an optimal balance between ET and RNC, achieving the lowest RNC in a shorter execution time.

Table 1: Performance Comparison of LOMAC and Baseline Methods on ER Graphs with 40, 200, and 1000 nodes. RNC values are reported as mean±standard deviation over 5 runs with different random seeds.

| Nodes | 40 | | | 200 | | | 1000 | | |
|---|---|---|---|---|---|---|---|---|---|
| | MR | RNC | ET | MR | RNC | ET | MR | RNC | ET |
| Tabu | 0.99 | 4.0±0.1 | 2.66 | 0.84 | 11.2±0.4 | 191.7 | NA | NA | NA |
| DLF-GA | 0.07 | 5.3±0.4 | 0.0078 | 0 | 14.0±0.2 | 0.01 | 0.34 | 47.7±0.9 | 0.15 |
| SAT-DRL | 0.99 | 4.0±0.1 | 25.23 | NA | NA | NA | NA | NA | NA |
| GNN-GCP | 0.77 | 4.2±0.4 | 0.5 | NA | NA | NA | NA | NA | NA |
| ECO-DQN | 0.95 | 4.1±0.4 | 0.08 | 0.8 | 11.2±0.4 | 1.51 | 0.62 | 45.6±0.9 | 77.7 |
| MCSS | 0.93 | 4.1±0.3 | 3.99 | NA | NA | NA | NA | NA | NA |
| LTMP | 0.9 | 4.1±0.3 | 0.009 | 0 | 13.0±0.2 | 0.15 | NA | NA | NA |
| AUTO-COT | 0.92 | 4.1±0.3 | 0.02 | 0.23 | 12.8±0.4 | 3.63 | NA | NA | NA |
| **LOMAC** | **0.99** | **4.0±0.1** | **0.37** | **0.84** | **11.2±0.3** | **5.24** | **0.78** | **42.8±0.8** | **43.1** |

## 5.2 PERFORMANCE ON REAL INSTANCES

To assess the effectiveness of LOMAC in real-world scenarios, we evaluated it on a small-scale COLOR02/03/04 Workshop dataset[2], which comprises instances with 11 to 149 vertices. Furthermore, we tested LOMAC on large-scale benchmark datasets, including Cora, Citeseer, and PubMed, with instances ranging from 2708 to 19717 vertices. Table 2 shows the performance of Tabu, GNN-GCP, SAT-DRL, LTMP, ECO-DQN, and LOMAC across these datasets in terms of RNC and ET.

---

[1]https://developers.google.com/optimization/cp

[2]https://mat.tepper.cmu.edu/COLOR02/

LOMAC consistently achieved or closely matched the minimal chromatic number in a shorter execution time. These results highlight LOMAC's robust generalization, efficiency, and clear superiority in both small-scale and large-scale real-world applications.

Table 2: Performance of LOMAC and Baseline Algorithms on COLOR02/03/04 Workshop dataset and large-scale benchmark datasets, Cora, Citeseer, and PubMed.

| Instance | Nodes | $\kappa$ | DLF-GA | | GNN-GCP | | SAT-DRL | | LTMP | | ECO-DQN | | LOMAC | |
|---|---|---|---|---|---|---|---|---|---|---|---|---|---|---|
| | | | RNC | ET | RNC | ET | RNC | ET | RNC | ET | RNC | ET | RNC | ET |
| myciel3 | 11 | 4 | 4 | 0.00025 | 4 | 0.567 | 4 | 1.25 | 4 | 0.0018 | 4 | 0.0075 | **4** | 0.26 |
| myciel4 | 23 | 5 | 5 | 0.0005 | 6 | 0.645 | 5 | 15.74 | 5 | 0.00126 | 5 | 0.0044 | **5** | 0.25 |
| myciel5 | 47 | 6 | 6 | 0.0013 | 7 | 0.808 | 6 | 293.58 | 4 | 0.0013 | 6 | 0.15 | **6** | 0.34 |
| huck | 74 | 11 | 11 | 0.0025 | 14 | 1.338 | NA | NA | 11 | 0.00324 | 11 | 0.26 | **11** | 0.51 |
| mugg100 25 | 100 | 4 | 4 | 0.004 | 3 | 0.904 | 4 | 9.38 | 4 | 0.00388 | 4 | 0.28 | **4** | 0.57 |
| games120 | 120 | 9 | 9 | 0.0058 | 13 | 1.481 | NA | NA | 9 | 0.00593 | 9 | 0.36 | **9** | 0.70 |
| anna | 138 | 11 | 12 | 0.00734 | 13 | 1.878 | NA | NA | 11 | 0.00794 | 11 | 0.47 | **11** | 0.74 |
| 2 - Insertions 4 | 149 | 4 | 5 | 0.0056 | 4 | 1.135 | NA | NA | 5 | 0.00828 | 5 | 0.45 | 5 | 0.77 |
| Cora | 2708 | 5 | 7 | 627 | NA | NA | NA | NA | NA | NA | 6 | 954 | **5** | 997 |
| Citeseer | 3327 | 6 | 9 | 854 | NA | NA | NA | NA | NA | NA | 7 | 1003 | **6** | 1020 |
| Pubmed | 19717 | 6 | 12 | 4632 | NA | NA | NA | NA | NA | NA | 10 | 4726 | 8 | 4929 |

## 5.3 MODEL ANALYSIS

**Complexity Analysis**. The dominant computational complexity of the proposed LOMAC method lies in the pseudonode-based GNN model during the testing process. For each coloring node, the computational cost consists of two parts: the GNN model to select an optimal coloring node with a linear time complexity of $O(N)$ for a graph of size $N$ (Sun et al., 2024), and the color assignment to color the node and update the node state of neighbors with a time complexity of $O(1)$. Due to the invalid action penalty mechanism, the occurrence of invalid actions (i.e. repeatedly selecting already colored nodes) is rare during testing. Therefore, the overall complexity of the LOMAC method is $O(N^2)$.

**Ablation Study**. To evaluate the contribution of different components, we design three ablation variants of LOMAC: (1) Ablation on Pseudo Nodes, which removes pseudo nodes from the message-passing network; (2) Ablation on Markov chain, which replaces the chain-based state transitions with direct color prediction using a predefined color set; and (3) Ablation on Potential Function, which removes the graph state potential-based reward function. As shown in Fig. 5, removing the Markov chain results in the largest performance drop in RNC, highlighting its crucial role in reducing the number of colors required. Excluding the potential function weakens the reward guidance, leading to less compact colorings, while removing pseudonodes reduces the efficiency of message passing. These results show that each component is essential and contributes to LOMAC's overall performance. Detailed results of these ablation experiments are provided in Appendix D.3.

## 6 CONCLUSION

In this paper, we introduce LOMAC, a novel GNN-based DRL framework for solving GCP. By integrating a one-way, two-dimensional Markov chain with a pseudonode-enhanced GNN, LOMAC significantly reduces the state space and computational complexity compared to traditional DRL approaches. We propose two key concepts, the Markov state potential and the graph state potential, and demonstrate their effectiveness in guiding the search for optimal solutions. Experimental results show that LOMAC outperforms existing methods on both synthetic and real-world datasets, achieving superior performance in terms of the number of colors required, matching ratio, and execution time. LOMAC also demonstrates strong generalization and efficiency across different types and sizes of graphs. Future work could explore applying LOMAC to other combinatorial optimization problems, further extending the applicability of GNN-based DRL models with one-way Markov chains.

## 7 ETHICS STATEMENT

This work focuses on the development of reinforcement learning and graph neural network methods to solve the graph coloring problem. The study does not involve human subjects, personal or sensitive data, or applications that could directly cause harm. All datasets used are publicly available benchmark graphs, and all external code or data strictly follow their respective licenses. We believe that this work raises no ethical concerns related to privacy, security, fairness, or potential misuse.

## 8 REPRODUCIBILITY STATEMENT

All code implementations of the LOMAC model and baseline methods are available in the Supplementary Material. The code is organized in a modular fashion with a clear separation between the definitions of the models, the training procedures, and the evaluation scripts. All experimental parameters, including learning rates, batch sizes, and network architectures, are explicitly specified in the training configuration files. The graph datasets used in our experiments, including both synthetic graphs and public benchmarks, are described in detail in the paper, with generation parameters provided. We have also included utility scripts for data preprocessing and result visualization to facilitate the reproduction of all figures and tables presented.

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

## LLM Usage Disclosure

Large language models (LLMs) were used in this work as writing and editing aids. The draft text for several sections, ablation study descriptions, and parts of the experimental analysis were initially generated or refined with LLM assistance, and then carefully reviewed, corrected, and finalized by the authors. The authors are solely responsible for the accuracy of all statements, the correctness of the code, and the validity of the results.

**Human verification & responsibility.** All claims, equations, proofs, references, and methodological descriptions were thoroughly checked by the authors. All experiments were independently rerun from clean environments, and all plots and tables were regenerated from verified outputs.

**Confidentiality & ethics.** No confidential or third-party material was provided to any LLM. We did not include hidden prompt injection text in the submission. All external data and code used in this work comply with their respective licenses, and all references were verified to correspond to real and relevant sources.

This disclosure is also reflected in the submission form as required by the ICLR policy.

## A    Preliminaries

This section provides an overview of deep Q-learning and message-passing neural networks, which are foundational to our study.

### A.1    Deep Q-learning

Deep Q-learning Mnih et al. (2015), a reinforcement learning technique, employs a trial-and-error approach within an uncertain environment to sequence decisions and actions toward a solution. Updates a Q-table to learn the reward associated with each state action pair, with the aim of choosing the state action pair that maximizes the reward. This method formalizes the decision making process as a Markov Decision Process (MDP), represented by the tuple $(\mathcal{S}, \mathcal{A}, \mathcal{R}, \mathcal{V}, \gamma)$, where $\mathcal{S}$ denotes the state space, $\mathcal{A}$ the action space, $\mathcal{R}$ the reward function, $\mathcal{V}(s)$ the value of being in state $s$ and $\gamma$ the discount factor for future rewards. The Bellman equation for Q-learning is expressed as:

$$\mathcal{V}(s) = \max_a [\mathcal{R}(s, a) + \gamma \mathcal{V}(s')] \tag{17}$$

Here, $s$ and $s'$ represent the current and subsequent states after taking action $a$. The reward value $Q(s, a)$ for action $a$ in state $s$ is updated based on:

$$Q(s, a) = Q(s, a) + \alpha[\mathcal{R} + (1 - D)\gamma \max_{a'} Q(s', a') - Q(s, a)] \tag{18}$$

with $\alpha$ as the learning rate, $s'$ and $a'$ denoting potential next states and actions, and $D$ indicating if the terminal state is reached ($D = 1$ for terminal states, otherwise $D = 0$).

## A.2 MESSAGE-PASSING NEURAL NETWORKS

We employ a message-passing neural network (MPNN) framework to enhance deep Q-learning for graph coloring, adept at processing graph-structured data Gilmer et al. (2017). The graph embedding transforms each vertex $v \in V$ into a multidimensional vector $h_v$. MPNN operates through a message passing phase and a read-out phase in $T$ steps, utilizing message functions $F$ and vertex update functions $U$. The message $f_v^{t+1}$ and the hidden state $h_v^{t+1}$ for the vertex $v$ are updated as follows:

$$f^{t+1}v = F_n(h_v^t, h_u^t u \in o(v)) \tag{19}$$

$$h_v^{t+1} = U_t(h_v^t, f_v^{t+1}) \tag{20}$$

Here, $o(v)$ denotes the set of neighbors for $v$, with $f^{t+1}v$ and $h_v^0 = s_v$ representing the message and initial state, respectively. Through this iterative process, vertex embeddings aggregate information from their neighborhood. The final read-out phase applies a function $R$ to the final embeddings, displaying action predictions (Q-values) as:

$$Q(s, a) = R(\{h_v^T\}_{a \in A}) \tag{21}$$

## B THE PROOF OF THEOREM 1

1) When all $N$ vertices are colored, $V''(\zeta_G) = 1$. According to Eq. (3), we have $\mathcal{V}(s_1) = V'_{i,N} + 1$, $\mathcal{V}(s_2) = V'_{i',N} + 1$. According to Eq. (2), $V'_{i,N} < V'_{i',N}$ since $i' < i$. Proposition 1 holds.

2) According to Eq. (2), $V'_{i,j+1} - V'_{i,j} \geq 1$. The range of $V''(.)$ is limited to [0, 1], where $V''(.)$ is the proportion of colored edges in all edges as defined in Eq. (3).

$$\begin{aligned}
\mathcal{V}(s(i, j+1)) &= V'_{i,j+1} + V''(\zeta_{G_{(i,j+1)}}) \\
&\geq V'_{i,j} + 1 \\
&\geq V'_{i,j} + V''(\zeta_{G_{(i,j)}}) \\
&\geq \mathcal{V}(s_{(i,j)})
\end{aligned} \tag{22}$$

The inequality $\mathcal{V}(s(i,j)) \geq \mathcal{V}(s(i, j+1))$ holds.

3) According to Eq. (2), $V'_{i,j} = V'_{i+1,j+1}$. For a newly selected vertex $u$, $u$ will transform adjacent uncolored edges of $u$ into colored edges. Without loss of generality, assume that $m''$ is the number of adjacent uncolored edges of $u$. Thus, for the vertex $u$, $u$ will transform adjacent uncolored edges of $u$ into colored edges. Without loss of generality, assume that $m''$ is the number of adjacent uncolored edges of $u$. Thus,

$$\begin{aligned}
\mathcal{V}(s(i, j)) &= V'_{i,j} + V''(\zeta_{G_{(i,j)}}) \\
&= V'_{i+1,j+1} + \frac{\zeta_{G_{(i,j)}}}{\|E\|} \\
&\leq V'_{i+1,j+1} + \frac{\zeta_{G_{(i,j)}} + m''}{\|E\|} \\
&= V'_{i+1,j+1} + \frac{\zeta_{G_{(i+1,j+1)}}}{\|E\|} \\
&= \mathcal{V}(s(i+1, j+1))
\end{aligned} \tag{23}$$

4) The maximum number of colored edges $\zeta_G$ in the graph $G$ is equal to the number of edges $\|E\|$. Thus, $V''(\zeta_G) = \frac{\zeta_G}{\|E\|} \leq 1$.

$$\begin{aligned}
\mathcal{V}(s(i+1, j)) &= V'_{i+1,j} + V''(\zeta_{G_{(i+1,j)}}) \\
&\leq V'_{i,j-1} + 1
\end{aligned} \tag{24}$$

$$\begin{aligned}
\mathcal{V}(s(i, j)) &= V'_{i,j} + V''(\zeta_{G_{(i,j)}}) \\
&\geq V'_{i,j-1} + 1 + V''(\zeta_{G_{(i,j)}}) \\
&> V'_{i,j-1} + 1
\end{aligned} \tag{25}$$

Thus, we have $\mathcal{V}(s(i,j)) > \mathcal{V}(s(i+1, j))$.

## C    The proof of Theorem 2

Assume that $k \neq \kappa$. The following two cases are taken into account.

1) $k > \kappa$. According to Theorem 1, $\mathcal{V}(s(k, N)) < \mathcal{V}(s(\kappa, N))$. It contradicts the notion that $\mathcal{V}(s_{(k,N)})$ is the largest potential value of the graph states.

2) $k < \kappa$. It contradicts the definition of the chromatic number, which is the smallest number of colors required for graph coloring.

Therefore, $k$ equals $\kappa$ when reaching a maximum on the potential of the graph states $\mathcal{V}(s(k, N))$.

## D    Details on Experiments

We conducted extensive experiments to evaluate the proposed LOMAC framework and reproduced the baseline models under consistent conditions. Hyper-parameters were determined via grid search based on validation loss, as summarized in Tab 3. All learnable parameters in LOMAC, including the weights of linear transformations, proximity measurement, and pseudonode states, were optimized jointly during training. For optimization, we adopted the Adam optimizer with a dynamically adjusted learning rate schedule.

### D.1    Descriptions of Baseline Algorithms

For graph coloring, we consider the following four types of baseline methods for performance comparison:

#### D.1.1    Traditional Heuristic Algorithms

**Tabu** (Blochliger & Zufferey, 2008): performs neighborhood search based on tabu lists and aspiration criteria.
**DLF-GA** (Gebremedhin et al., 2013): assigns the smallest available color sequentially through local optimization.

#### D.1.2    GNN Methods

**GNN-GCP** (Lemos et al., 2019): uses GNN message passing to update the embeddings, predicts the colorability of $C$ through supervised learning and generates solutions.

#### D.1.3    GNN-based DRL Algorithms

**SAT-DRL** (Yolcu & Póczos, 2019): encodes graph coloring as a CNF formula, models variable relationships using GNN, and optimizes variable selection through reinforcement learning.
**ECO-DQN** (Barrett et al., 2020): explores the solution space dynamically through reinforcement learning and optimizes vertex states with a reward mechanism.
**MCSS** (Yuan et al., 2024): selects optimal column combinations dynamically using neural networks and a multicolumn selection strategy driven by reinforcement learning.

#### D.1.4    LLMs

**LTMP** (Zhou et al., 2023): solves complex problems by sequentially addressing simpler subproblems that depend on the solutions of previous ones.
**AUTO-COT** (Zhang et al., 2023): clusters various problems and uses LLM to generate reasoning chains for the construction of automatic demonstrations.

### D.2    Experimental setup.

The experiments were carried out using an RTX 4090 GPU and an Intel(R) Xeon(R) Platinum 8474 CPU, with software implementation in PyTorch 2.5.1. Due to limitations in memory resources, we

limit the experience replay buffer to 5000 samples. The GNN employs the Adam optimizer with a learning rate $\rho$, dynamically adjusted as:

$$\rho = \begin{cases} \frac{0.001\tau}{1000} & 0 \leq \tau \leq 1000 \\ 0.001 - 0.00095\frac{\tau-1000}{19000} & 1000 < \tau \leq 20000 \\ 0.00005 & \tau > 20000 \end{cases} \quad (26)$$

Table 3: Hyper-parameter setups for LOMAC.

| DATASET | #MESSAGE STEPS $(T)$ | HIDDEN DIM. | Q-SPACE DIM. | #Q-UNITS $(n_q)$ | #PSEUDO NODES $(n_p)$ | DROPOUT |
|---|---|---|---|---|---|---|
| ER-20 | 2 | 128 | 64 | 20 | 8 | 0.1 |
| ER-40 | 2 | 128 | 64 | 40 | 20 | 0.1 |
| ER-60 | 2 | 128 | 64 | 60 | 30 | 0.1 |
| ER-200 | 2 | 128 | 64 | 200 | 96 | 0.2 |
| ER-500 | 3 | 128 | 64 | 500 | 256 | 0.2 |
| ER-1000 | 3 | 128 | 64 | 1000 | 520 | 0.3 |
| BA-20 | 2 | 128 | 64 | 20 | 8 | 0.1 |
| BA-40 | 2 | 128 | 64 | 40 | 20 | 0.1 |
| BA-60 | 2 | 128 | 64 | 60 | 30 | 0.1 |
| BA-200 | 2 | 128 | 64 | 200 | 96 | 0.2 |
| CORA | 4 | 128 | 64 | 2708 | 1256 | 0.3 |
| CITESEER | 4 | 128 | 64 | 3327 | 1256 | 0.3 |
| PUBMED | 8 | 128 | 64 | 19717 | 5200 | 0.3 |

## D.3 ABLATION STUDY

We have demonstrated the effectiveness of LOMAC in solving GCP. To further analyze the contribution of individual components, we conduct three ablation studies, with results shown in Fig. 5.

**Ablation Study on the Markov Chain** To assess the impact of the one-way Markov chain, we performed an ablation study by removing this component and allowing the model to predict the color of each node directly. In this setup, with a fixed number of colors $K$, we increment $K$ if a feasible coloring cannot be achieved. Without chain-based state transitions, the model relies on a predefined color space, which increases the solution complexity and reduces the guidance from structured state evolution. The results show that removing the Markov chain increases the number of required colors, highlighting its effectiveness in guiding compact colorings.

**Ablation Study on the Potential Function** To evaluate the effect of the potential-based reward of the graph state, we performed an ablation study by removing this component and adopting a simple potential-free reward. In this setup, the agent receives a reward of $+1$ for reusing an existing color and keeping the color valid. It gets $0$ when a new color is introduced, and a penalty of $-\lambda$ for invalid actions. The results show that removing the potential function removes strong incentives for compact colorings, resulting in an increased number of required colors.

**Ablation Study on Pseudo Nodes** To evaluate the impact of pseudonode-enhanced message passing, we replace the graph neural decision network with a basic GNN. In this setup, message passing relies solely on the original graph topology, limiting efficiency and overall performance. The pseudonode-based message-passing mechanism embeds both graph nodes and pseudonodes into a unified latent space, enabling more flexible message-passing and reducing dependency on the graph topology. The results confirm the effectiveness of this mechanism. These three ablation studies isolate the contribution of each key component in LOMAC.

## D.4 STATISTICAL SIGNIFICANCE ANALYSIS

To assess the robustness of the proposed LOMAC method in different sizes of graphs, we performed pairwise Wilcoxon signed rank tests between LOMAC and baseline algorithms on ER graphs with

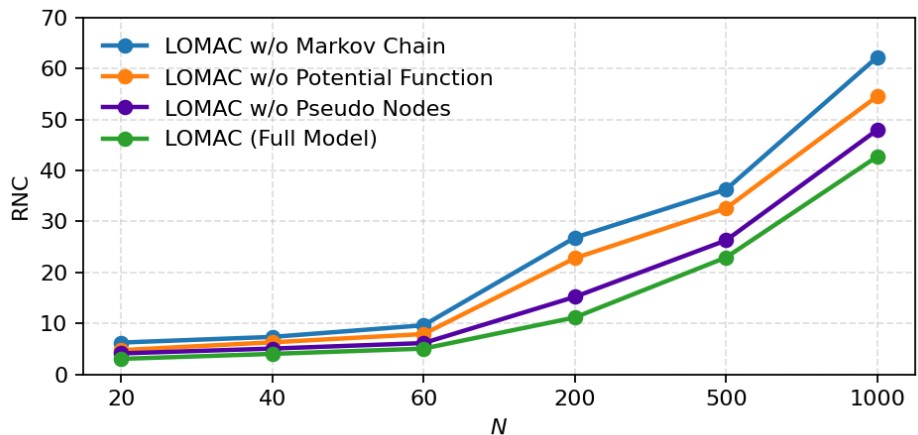

Figure 5: Ablation on the modules of LOMAC.

$N = 20, 40, 60, 200, 500$ and $1000$ nodes. We used a significance level of $p < 0.05$. The results are summarized in Table 4, where a smaller p-value provides stronger evidence that LOMAC significantly outperforms the baseline algorithms in terms of RNC.

Table 4: Wilcoxon signed-rank test results on ER graphs with different node sizes. LOMAC shows statistically significant improvements over all baselines in terms of the number of colors.

| Ours | Baseline | p-value (RNC) | | | | | |
|---|---|---|---|---|---|---|---|
| | Nodes | 20 | 40 | 60 | 200 | 500 | 1000 |
| LOMAC | Tabu | 0.042 | 0.037 | 0.041 | 0.039 | < 0.001 | < 0.001 |
| LOMAC | DLF-GA | < 0.001 | < 0.001 | < 0.001 | < 0.001 | < 0.001 | < 0.001 |
| LOMAC | SAT-DRL | 0.021 | 0.019 | 0.023 | < 0.001 | < 0.001 | < 0.001 |
| LOMAC | GNN-GCP | 0.011 | 0.009 | < 0.001 | < 0.001 | < 0.001 | < 0.001 |
| LOMAC | ECO-DQN | 0.017 | 0.016 | 0.021 | 0.015 | < 0.001 | < 0.001 |
| LOMAC | MCSS | 0.024 | 0.023 | 0.025 | < 0.001 | < 0.001 | < 0.001 |
| LOMAC | LTMP | 0.012 | 0.011 | < 0.001 | < 0.001 | < 0.001 | < 0.001 |
| LOMAC | AUTO-COT | 0.021 | 0.032 | 0.018 | < 0.001 | < 0.001 | < 0.001 |

### D.5 ADDITIONAL RESULTS ON SYNTHETIC AND REAL-WORLD INSTANCES

We provide additional results on both synthetic graphs (ER and BA) and real-world benchmark instances. Tables 5 and 6 report performance on ER and BA graphs with 20, 60, and 500 nodes. Across different sizes, LOMAC achieves the lowest RNC in the shortest execution time.

We also include results on selected real-world instances in Table 6. LOMAC consistently reaches optimal colorings with manageable run-time, highlighting its robustness and efficiency across diverse graph types and scales.

Table 5: Performance Comparison of LOMAC and Baseline Methods on ER Graphs with 20, 60, and 500 nodes. RNC values are reported as mean±standard deviation over 5 runs with different random seeds.

| Nodes | 20 | | | 60 | | | 500 | | |
|---|---|---|---|---|---|---|---|---|---|
| | MR | RNC | ET | MR | RNC | ET | MR | RNC | ET |
| Tabu | **1** | 3.0±0.1 | 0.79 | 0.96 | 5.0±0.2 | 20.61 | NA | NA | NA |
| DLF-GA | 0.34 | 3.7±0.4 | 0.0001 | 0 | 6.8±0.4 | 0.0016 | 0.2 | 26.3±0.6 | 1.93 |
| SAT-DRL | **1** | 3.0±0.1 | 3.01 | 0.9 | 5.1±0.3 | 90.5 | NA | NA | NA |
| GNN-GCP | **1** | 3.0±0.1 | 0.48 | 0.71 | 5.1±0.4 | 0.547 | NA | NA | NA |
| ECO-DQN | 0.93 | 3.1±0.3 | 0.02 | 0.87 | 5.1±0.3 | 0.12 | 0.74 | 25.4±0.8 | 23.8 |
| MCSS | 0.9 | 3.1±0.3 | 0.79 | 0.92 | 5.1±0.3 | 3.54 | NA | NA | NA |
| LTMP | 0.9 | 3.13±0.3 | 0.0028 | 0.5 | 5.1±0.5 | 0.02 | NA | NA | NA |
| AUTO-COT | 0.8 | 3.23±0.4 | 0.01 | 0.44 | 5.55±0.5 | 0.04 | NA | NA | NA |
| LOMAC | **1** | **3.03±0.1** | **0.16** | **0.96** | **5.04±0.2** | **0.84** | **0.82** | **22.94±0.6** | **7.21** |

Table 6: Performance Comparison of LOMAC and Baseline Methods on BA Graphs with 20, 60, and 500 nodes, in terms of MR, RNC, and ET. RNC values are reported as mean±standard deviation over 5 runs with different random seeds.

| Nodes | 20 | | | 60 | | | 500 | | |
|---|---|---|---|---|---|---|---|---|---|
| | MR | RNC | ET | MR | RNC | ET | MR | RNC | ET |
| Tabu | 0.99 | 4.9±0.37 | 1.21 | 0.95 | 5.04±0.2 | 26.56 | NA | NA | NA |
| DLF-GA | 0.05 | 6.41±0.46 | 0.01 | 0 | 8.11±0.3 | 0.03 | 0 | 13.24±0.4 | 0.28 |
| SAT-DRL | 1 | 4.89±0.31 | 34.7 | 0.81 | 5.2±0.4 | 84.78 | NA | NA | NA |
| GNN-GCP | 0.98 | 4.91±0.42 | 0.35 | 0.80 | 5.32±0.5 | 0.546 | NA | NA | NA |
| ECO-DQN | 0.96 | 4.93±0.46 | 0.03 | 0.95 | 5.04±0.2 | 0.21 | 0 | 9.24±0.4 | 7.24 |
| MCSS | 0.99 | 4.9±0.31 | 0.75 | 0.77 | 5.21±0.4 | 4.67 | NA | NA | NA |
| LTMP | 0.94 | 4.95±0.31 | 0.0049 | 0.81 | 5.18±0.2 | 0.013 | NA | NA | NA |
| AUTO-COT | 0.97 | 4.92±0.31 | 0.07 | 0.89 | 5.1±0.3 | 0.32 | NA | NA | NA |
| LOMAC | **1** | **4.89±0.31** | **0.28** | **0.95** | **5.04±0.19** | **1.61** | **0.72** | **7.28±0.32** | **9.26** |

Table 7: Performance of LOMAC and Baseline Algorithms on COLOR02/03/04.

| Instance | Nodes | $\chi_0$ | DLF-GA | | GNN-GCP | | SAT-DRL | | LTMP | | ECO-DQN | | LOMAC | |
|---|---|---|---|---|---|---|---|---|---|---|---|---|---|---|
| | | | RNC | ET | RNC | ET | RNC | ET | RNC | ET | RNC | ET | RNC | ET |
| queen5_5 | 25 | 5 | 8 | 0.00061 | 6 | 0.662 | 5 | 38 | 6 | 0.00038 | 5 | 0.05 | **5** | 0.25 |
| 3 - Insertions 3 | 56 | 4 | 4 | 0.0015 | 4 | 0.703 | 4 | 9.19 | 4 | 0.00138 | 4 | 0.12 | **4** | 0.39 |
| david | 87 | 11 | 12 | 0.0033 | 14 | 1.421 | NA | NA | 12 | 0.00362 | 13 | 0.29 | **11** | 0.53 |
| mugg88 25 | 88 | 4 | 4 | 0.0033 | 4 | 0.837 | 4 | 9.82 | 4 | 0.00356 | 4 | 0.26 | **4** | 0.52 |
| mugg100 1 | 100 | 4 | 4 | 0.0045 | 2 | 0.899 | 4 | 11.17 | 4 | 0.00402 | 4 | 0.27 | **4** | 0.57 |

# E   ALGORITHM DESCRIPTION

**Algorithm 1** The training process of LOMAC

1: **In**: Randomly generated $W$ graph samples, $p$, $K$, $b$, $z$
2: **Out**: The network parameters $\theta$. /* $\theta'$ indicates the target network parameters
3: Initialize the network with random $\theta$.
4: $\tau = 0$. /* $\tau$ indicates the current step number
5: **for** each graph sample **do**
6:    $V^{cv} = \varnothing, \mathcal{C} = \varnothing$
7:    Update the value of $\varepsilon$
8:    **for** $i = 1$ **to** $K$ **do**
9:      **if** $V^{cv} \neq V$ **then**
10:        $r = r + 1$
11:        $\varepsilon'$=random(0,1). /* get a random number in [0,1]
12:        $v^* = \begin{cases} \varepsilon' < \varepsilon : \text{Choose random } v^* \in V^{cv} \\ \varepsilon' \geq \varepsilon : \textbf{The GNN-based decision-making phase,} \\ \text{argmax}_{a \in A} Q(s,a) \end{cases}$
13:        $a^* = v^*$
14:        **The color assignment phase:**
15:        **if** $C^{v^*} \neq \varnothing$ **then**
16:           stochastically assign a color from $C^{v^*}$ to $v^*$
17:           $R(s(i,j), a^*) = \mathcal{V}(s(i, j+1)) - \mathcal{V}(s(i,j))$
18:           Update the attribute values of $s(i, j+1)$
19:           $V^{cv} = V^{cv} \cup v^*, s^c = s(i, j+1)$
20:        **else**
21:          **if** $C^{v^*} = \varnothing$ **then**
22:            assign a new color $c^{\text{new}}$ to $v^*$
23:            $R(s(i,j), a^*) = \mathcal{V}(s_{(i+1, j+1)}) - \mathcal{V}(s(i,j))$
24:            Update the attribute values of $s(i+1, j+1)$
25:            $V^{cv} = V^{cv} \cup v^*, \mathcal{C} = \mathcal{C} \cup c^{\text{new}}, s^c = s(i+1, j+1)$
26:          **else**
27:            $R(s(i,j), a^*) = z, s^c = s(i,j)$
28:          **end if**
29:        **end if**
30:        Update $Q(s_{(i,j)}, a^*)$ according to Eq. (18)
31:        **The prioritized experience replay phase:**
32:        Compute the value of $p$ according to Eqs. (7) and (8)
33:        Add $[s(i,j), a^*, Q(s(i,j), a^*), s^c, p]$ into the prioritized experience replay buffer.
34:        **if** $\tau \bmod p == 0$ **then**
35:          Get $b$ random samples $B$ from the buffer
36:          learn $\theta$ given training samples $B$.
37:        **end if**
38:      **end if**
39:    **end for**
40: **end for**

**Algorithm 2** The testing process of LOMAC

1: **In**: $G = (V, E)$, the network with parameters $\theta$
2: **Out**: $C, S$
3: $V^{cv} = \varnothing, \mathcal{C} = \varnothing$
4: **for** $i = 1$ **to** $K$ **do**
5:    **if** $V^{cv} \neq V$ **then**
6:       **The GNN-based decision-making phase:**
7:       $a^* = v^* = \text{argmax}_{a \in A} Q(s, a)$
8:       **The color assignment phase:**
9:       **if** $C^{v^*} \neq \varnothing$ **then**
10:          stochastically assign a color from $C^{v^*}$ to $v^*$
11:          $R(s(i, j), a^*) = \mathcal{V}(s(i, j + 1)) - \mathcal{V}(s(i, j))$
12:          Update the attribute values of $s(i, j + 1)$
13:          $V^{cv} = V^{cv} \cup v^*, s^c = s(i, j + 1)$
14:       **else**
15:          **if** $C^{v^*} = \varnothing$ **then**
16:             assign a new color $c^{\text{new}}$ to $v^*$
17:             $R(s(i, j), a^*) = \mathcal{V}(s_{(i+1, j+1)}) - \mathcal{V}(s(i, j))$
18:             Update the attribute values of $s_{(i+1, j+1)}$
19:             $V^{cv} = V^{cv} \cup v^*, \mathcal{C} = \mathcal{C} \cup c^{\text{new}}, s^c = s(i + 1, j + 1)$
20:          **else**
21:             $R(s(i, j), a^*) = z, s^c = s(i, j)$
22:          **end if**
23:       **end if**
24:    Update $Q(s(i, j), a^*)$ according to Eq. (18)
25:    **end if**
26: **end for**

