# OpenReview forum: "LOMAC: GNN-based Deep Reinforcement Learning with One-Way Markov Chain for Graph Coloring"
_ICLR.cc/2026/Conference — ICLR 2026 Conference Withdrawn Submission_

### Official Review · Reviewer_Lg7N · 2025-10-22

**Soundness:** 3
**Presentation:** 1
**Contribution:** 2
**Rating:** 2
**Confidence:** 4

**Summary:**

This paper introduces LOMAC, a deep reinforcement learning method for solving the graph colouring problem using a GNN architecture. The main contribution of the paper is to simplify the problem representation using an abstraction to a Markov chain which encompasses multiple equivalent states in the underlying partial graph colouring. The proposed model achieves a statistically significant performance improvement on large graphs compared to the included baselines.

**Strengths:**

S1. The approach of reducing the state space to collapse symmetrical colourings is wise and well-motivated.

**Weaknesses:**

The main weaknesses of the paper are around its clarity. The obtained results are reasonable but given the lack of clarity it is currently impossible to understand how the method works, judge whether it is techically correct, or build an intuition as to why it performs any better than the alternatives.

W1. The MDP definition is wrong. The transition function is not given at all. The state value function $\mathcal{V}(s)$ is not part of the MDP model, and would instead be derived from the reward function. In fact, if we know the state-value function, it is not necessary to use a reward function or any planning algorithms at all, because the action is simply chosen that leads to the state maximising $\mathcal{V}$.

W2. Action selection is not sufficiently explained: while it is clear that a node is selected to be coloured, it is not explained how the colour is chosen, and it is not always as simple as choosing one non-conflicting colour. It is possible to construct a graph in which there might be multiple valid colours to choose for a particular node, but that choice of colour impacts later choices, with a poor choice forcing an extra colour to be used to colour later nodes. The paper does not explain how colours are assigned in such a scenario, nor does it mention the ramifications of the choice. This links to W1, as a clear transition function is required to understand what the actions do.

W3. The purpose and origin of Markov state potentials and graph state potentials is not clear.

W4. The motivation and purpose of the pseudonodes is not explained. If this is a novel construction it needs much more explanation. If it is not, it needs references to prior works using this architecture.

**Questions:**

C1. Can you explain why you chose to use a penalty term for invalid actions instead of using action masking?

C2. Are there any DRL papers addressing the GCP that use any kind of representation that ignores equivalent colourings, or is this the first?

C3. How is the potential of a Markov state defined? Are the inequalities in Eq. 2 defined or derived? If they are defined, what is the justification for the choices and the specific numbers? Are they just used to construct the reward function?

C4. In Table 2 there are entries for RNC which are smaller than $\kappa$ for GNN-GCP and LTMP. Are these typos, or is there some explanation for how this can be?

C5. How was the set of six attributes for the nodes chosen, and are there any ablations that demonstrate their necessity?

C6. How did you arrive at the design for the pseudonode adaptation?

C7. How is $M$ chosen for the number of pseudonodes?

C8. Can you explain why LTMP is so fast on the instances above the line in Table 2 but did not produce solutions for those below the line?

### Minor comments
- Figure 1 is too small to read easily and the top two panels are difficult to understand.
- Typo in 3.2: "the action $a_{v_j}$, i.e., selecting and coloring $v_i$" -> should be $v_j$.
- In Eq. 4 the notation $s(i,j)$ has not been introduced, assuming $s$ is a graph state.
- In Tables 1, 2, 5, 6, and 7 there are a number of entries with the same performance as your method which have not been bolded. I would recommend placing all best performances in bold font, and consider placing second place in italics.

---

### Official Review · Reviewer_4iFv · 2025-10-24

**Soundness:** 3
**Presentation:** 2
**Contribution:** 2
**Rating:** 2
**Confidence:** 4

**Summary:**

This paper uses GNNs and RL to solve graph coloring problems. They introduce LOMAC which reduces the Markov states  from $\mathcal{O}(K^{|V|})$  to $\mathcal{O}(|V|^2)$. Experimentally, the method achieves strong results on multiple datasets of different sizes and from different domains.

**Strengths:**

- **(S1) Contribution, Originality:** The main contribution of reducing the number of Markov states from $\mathcal{O}(K^{|V|})$ to $\mathcal{O}(|V|^2)$ is elegant.
- **(S2) Clarity:** In general, the paper is well written and easy to follow.
- **(S3) Significance:** The experiments compare against a diverse set of base lines on a diverse of datasets. Especially for large graphs the proposed method achieves good results: near optimal colorings at a fast runtime.

**Weaknesses:**

Note that my background is not RL. Thus, I am not qualified to judge the qualities of the RL contributions of this paper.

- **(W1) Writing:** There are numerous tiny mistakes in this paper that making reading it more difficult than it should be (see Miscellaneous below). While fixing them is definitely possible, this shows that this paper would have required more time for polish.
- **(W2) Experiments:** The presentation of experimental results is misleading and makes the proposed method sound better than the experiments demonstrate.
	- Results in Table 1 and 2 are boldened without describing what bold means. This is misleading as bold is usually used to mark the best result in a table whereas here that is not the case (e.g. in Table 1 for 40 nodes multiple models achieve a matching ratio of 0.99 but only LOMAC is boldened).
	- Table 1: LOMAC only convincingly outperforms the baseline in the setting with 1000 nodes. Below, ECO-DQN achieves comparable MR and RNC while being 3-5 times faster. It would be more convincing if the authors could present more results on large graphs as this seems to be more aligned with the strengths of LOMAC.
	- Table 2: On the smaller graphs (from the workshop database), LOMAC is again convincingly beaten by ECO-DQN. While both models achieve optimal solutions in almost all cases, ECO-DQN is 1.5-50 times faster. Again, only on the larger datasets (Cora, Citeseer, Pubmed) does LOMAC convincingly beat all other models, implying that maybe the experiments should be focused more on large-scale datasets.
- **(W3) Code:** While the authors provide code, it is not documented at all. There are no instructions on how to set up the environments and on how to reproduce the results.
- **(W4) Contribution:** Overall, while the core idea is elegant (S1), it is also straight-forward and combined with standard GNN and RL methods. Thus, I think the contribution is still too small for publication.

**Questions:**

- What does LOMAC mean?

### Miscellaneous
- You use the term "chromatic number" multiple times without defining it first.
- Figure 1 (centrally positioned on page 1) is very dense and difficult to parse.
- Citations without a reference to venue
- _"For example, Inaba et al. (Inaba et al., 2022) applied the Potts model to graph coloring, iteratively updating interaction matrices to minimize Potts energy."_
Please use `\citet{...}`, then you can use the citation to mention Inaba et al. in the main text.
- The edge definition $E=  \\{ (i,j) \mid i, j \in V\\}$  implies that the graph fully connected. I think you mean $E \subseteq \\{ (i,j) \mid i, j \in V\\}$.
- _"In this model, transitioning from state $m(i, j)$ involves three scenarios: 1) If a valid
color exists for vertex $v$, the state moves to $m(i, j + 1)$;"_
What does this mean exactly? In particular, what is vertex $v$? Is it any vertex, i.e., can we do case (1) as long as any additional vertex is color able without adding an additional color?
- _"Definition 2 (Colored Degree of a Graph).The ..."_ Missing space after the dot.
- The notation $V'_{i,j}$ is confusing when $V$ is the vertex set.
- References to tables and figures are not clickable (please use `\ref{...}`)
- Figure 5 is discussed in the main text but is actually deep in the appendix on page 16.
- Page 4 contains 8 definitions in different definition environments in a row. This is jarring to read. requires a lot of space. Furthermore, it is not clear for what these definitions are needed.

**Overall,** while I think that the idea of reducing the state space to speed up computation is interesting, this paper clearly not publication ready. It misses polish (in both the text and code documentation) and the experiments are not sufficiently set up to highlight the models' strengths. Thus, I vote to reject.

---

### Official Review · Reviewer_GB3U · 2025-10-30

**Soundness:** 2
**Presentation:** 3
**Contribution:** 2
**Rating:** 4
**Confidence:** 4

**Summary:**

This paper introduces LOMAC, a novel GNN-based deep reinforcement learning (DRL) framework designed to solve the Graph Coloring Problem (GCP). The authors propose a one-way, two-dimensional Markov chain integrated with a linear-complexity GNN model enhanced by pseudonodes for message passing. By reducing the state space from O(KN) to O(NK), the model improves both space and computational complexity. Additionally, the paper introduces concepts like Markov state potential and graph state potential to guide the optimization process. Experimental results demonstrate that LOMAC outperforms several baseline algorithms across various graph sizes, showing superior performance in terms of the number of colors required, matching ratio, and execution time.

**Strengths:**

1. The proposed one-way Markov chain for graph coloring significantly reduces computational complexity, a major strength of the paper. This innovation helps scale the solution to larger graphs efficiently.

2.The paper introduces Markov state potential and graph state potential and establishes theoretical results to guide the coloring process. This provides a solid theoretical basis for the proposed framework.

3.The experimental results demonstrate LOMAC's superior performance compared to multiple baseline methods, including traditional heuristics and GNN-based DRL algorithms. The paper clearly shows the efficiency of the model in both small- and large-scale graph instances, including real-world benchmarks like Cora, Citeseer, and PubMed.

**Weaknesses:**

1. Despite the model's reduction in state space, the computational complexity of LOMAC remains O(N²). While this is an improvement over traditional methods, it may still pose challenges for extremely large graphs. Further optimization in this area could be explored.

2.The paper focuses mainly on comparing LOMAC to traditional heuristic algorithms and a few baseline DRL methods. A more comprehensive comparison with other recent GNN-based DRL models (e.g., S2V-DQN or FastColorNet) would further strengthen the claim of LOMAC’s superiority.

3.The reward function that guides the model towards optimal solutions could benefit from a more detailed theoretical explanation. While the empirical results are promising, a deeper understanding of the reward shaping and how it impacts model training and convergence would add clarity.

**Questions:**

1. While LOMAC reduces the number of states, the overall complexity is still O(N²). How does the model perform with even larger-scale graphs (e.g., thousands of nodes)? Are there specific optimizations that can reduce the runtime for extremely large graphs?
2. Can LOMAC be extended to solve other combinatorial optimization problems beyond graph coloring? For instance, could it be applied to problems like Maximum Independent Set or Graph Partitioning?
3. How does LOMAC compare in terms of training time and convergence speed with other recent GNN-based DRL models for graph coloring, such as S2V-DQN or FastColorNet? Would a deeper comparison with these models reveal more about LOMAC’s strengths and weaknesses?
4.The paper discusses the pseudonode-enhanced GNN but doesn’t elaborate in detail on the impact of this enhancement. Could the authors provide more insights into how pseudonodes contribute to reducing computational overhead and improving message passing efficiency?

---

### Note · Authors · 2025-12-10

I have read and agree with the venue's withdrawal policy on behalf of myself and my co-authors.